# Dual-Energy Computed Tomography Applications to Reduce Metal Artifacts in Hip Prostheses: A Phantom Study

**DOI:** 10.3390/diagnostics13010050

**Published:** 2022-12-23

**Authors:** Daniele Conti, Fabio Baruffaldi, Paolo Erani, Anna Festa, Stefano Durante, Miriam Santoro

**Affiliations:** 1Medical Technology Laboratory, IRCCS Istituto Ortopedico Rizzoli, 40136 Bologna, Italy; 2Nursing, Technical and Rehabilitation Assistance Service, IRCCS Istituto Ortopedico Rizzoli, 40136 Bologna, Italy

**Keywords:** arthroplasty, diagnostic imaging, dual-energy computed tomography, hip prosthesis, metal artifact reduction, spectral imaging CT

## Abstract

Metal components of hip prostheses cause severe artifacts in CT images, influencing diagnostic accuracy. Metal artifact reduction (MAR) software and virtual monoenergetic reconstructions on dual-energy CT (DECT) systems are possible solutions that should be considered. In this study, we created a customized adjustable phantom to quantify the severity of artifacts on periprosthetic tissues (cortical and spongious bone, soft tissues) for hip prostheses. The severity of artifacts was classified by different thresholds of deviation from the CT numbers for reference objects not affected by artifacts. The in vitro setup was applied on four unilateral and three bilateral configurations of hip prostheses (made of titanium, cobalt, and stainless steel alloys) with a DECT system, changing the energy of virtual monoenergetic reconstructions, with and without MAR. The impact of these tools on the severity of artifacts was scored, looking for the best scan conditions for the different configurations. For titanium prostheses, the reconstruction at 110 keV, without MAR, always minimized the artifacts. For cobalt and stainless-steel prostheses, MAR should always be applied, while monoenergetic reconstruction alone did not show clear advantages. The available tools for reducing metal artifacts must therefore be applied depending on the examined prosthetic configuration.

## 1. Introduction

Computed tomography (CT) is frequently applied on patients with hip prosthesis. Unfortunately, the metal components of a hip prosthesis cause severe artifacts in CT images and interfere with diagnostic accuracy, especially in the interface between implant and bone or to the adjacent soft tissues. Metal artifacts appear as dark and bright streaks or bands, as they are caused by the combination of physical phenomena, among which the most important are beam hardening, photon starvation, Compton scattering, and partial volume effects [1,2]. The grade of metal artifacts depends on the size, the orientation, and the different metal alloys of the implants [3]. These characteristics differ according to the wide variety of hip prosthesis models, and the severity of metal artifacts grows in patients with a bilateral implants and with the increase in the atomic number of the metal [4].

Previous studies show a significant reduction of beam hardening artifacts using virtual monochromatic images obtained with dual-energy computed tomography (DECT) [5,6,7,8,9]. DECT does, in fact, make it possible to reconstruct simulated images as if they were acquired with a monochromatic X-ray beam because of the linear combination of the projection data from two energy sources, i.e., two different X-ray spectra obtained with low and high kVp settings. The effectiveness of DECT in metal artifact reduction and the impact of different metal alloys was investigated in several studies [10,11,12,13,14,15,16,17], which found that titanium implants give rise to fewer severe metal artifacts compared to metals with a higher atomic number, such as stainless steel, cobalt-chrome, and tantalum-based porous biomaterial. As different metal alloys result in different degrees of artifacts, it is reasonable that implant-specific CT imaging protocols are necessary to achieve optimal monochromatic energies (keV) suitable for limiting metal artifacts, and are more helpful than using generalized protocols for all implants [15].

Using high monochromatic energies (80–140 keV) [18] in small unilateral implants made from lighter metals, such as titanium, leads to consistent artifact reduction. Instead, with large hip implants, bilateral hip implants, and hip implants made from heavier metals (e.g., cobalt or chrome), the minimization of metal artifacts is limited. In these cases, the use of higher keVs might result in a further decrease in metal artifacts, although photon starvation and scatter still also affect the resulting image in high keV images [7].

To overcome this problem, CT manufacturers created proprietary software for metal artifact reduction (MAR), which can be used with conventional single-energy CT or DECT. The value and the benefits of these software programs were analysed in several studies with different clinical questions [10,19]. Moreover, the combined use of MAR software and virtual monochromatic images in DECT could help to further improve the diagnostic quality of these types of images [16].

Previous studies have evaluated the combination of both tools to reduce metal artifacts in DECT [20,21,22,23,24], although the efficacy of these combined approaches on different types of hip prosthesis materials remains to be demonstrated. In particular, attention must be given to the possible generation of new, MAR-related artifacts that affect larger metal components and heavier metals [16]. The optimal combination of virtual monochromatic images with or without MAR software should always be considered in DECT. Unfortunately, the literature does not always provide clear guidelines for this choice in relation to the specific prosthetic configuration.

One limitation of some of these studies is their experimental setup. In fact, most of these studies are performed on phantoms simulating the surrounding bone tissue and the prosthetic implant. The use of high-density materials in the phantom, only mimicking bone tissue, does not provide a satisfactory prediction of the impact of metal artifacts on middle and low-attenuation coefficient tissue [7]. Moreover, artifact measurements were taken in regions not directly in contact with the prosthetic components, therefore applying a less critical condition than the condition normally seen in clinical practice for periprosthetic tissues.

The aim of this study is to evaluate the combined use of MAR software and virtual monochromatic imaging available on a DECT system to identify the optimal implant specific DECT protocol for metal artifact minimization. We investigated four unilateral configurations and three bilateral configurations with Ti6Al4V, CoCrMo, and stainless steel components. The quantification and the optimization of metal artifact reduction were done using an experimental phantom with materials simulating compact bone, spongious bone, and soft tissue. By using an adjustable phantom with a fine positioning system capable of putting the measuring area for artifacts in direct contact with the prosthetic components, we obtained a more realistic simulation of the clinical scenario. Image quality for the different configurations was obtained from objective quantitative measurements, such as CT number stability, noise values, and signal-to-noise-ratios (SNRs). Then, the analysis of artifacts was summarized in a format alike a practical guideline for the best choice among MAR and virtual monochromatic imaging for the hip prostheses configurations studied.

## 2. Materials and Methods

### 2.1. Spectral CT Using Fast kV Switching Technique

The Revolution^TM^ Discovery^TM^ CT (GE Healthcare, Waukesha, WI, USA) DECT system, with software version 17BW50./B_SP1, consists of a single tube with the ability to rapidly switch between two different kV settings (80 kVp and 140 kVp) to obtain two different datasets. Specifically, the data are acquired twice for each projection by the gemstone spectral imaging (GSI) detector and are combined in the projection space. Through spectral separation, it is possible to obtain virtual monochromatic imaging (VMI) ranging from 40 to 140 keV.

### 2.2. Hip Phantom and Prostheses

The hip phantom was based on two polypropylene (PP) containers, with dimensions of 365 mm(L) × 245 mm(W) × 227 mm(H), filled with demineralized water. The two containers were placed side by side to simulate the left and right sides of the pelvis. Inside each container, we arranged different left and right prostheses, mimicking unilateral and bilateral hip arthroplasties within the human body, as reported in Table 1. The prostheses were positioned using LEGO bricks and plates (The LEGO^®^ Group, Billund, Denmark), made with low X-ray attenuating plastics (ABS, High Impact Polystyrene, PE). A LEGO baseplate was attached to the inferior base of each container. Each prosthesis was inserted into the container and supported with LEGO elements and fastened with rubber bands to prevent movements during the scan. To evaluate artifacts as closely as possible to the prosthesis (distance less than 0.5 mm), multi-material pellets were assembled. The prostheses were surrounded by “triplets”, each consisting of three cylindrical pellets made of different materials. The materials used for the pellets (the diameter of each cylinder was 10 mm), produced by Manifattura Cattaneo S.p.A., Samarate (Varese), Italy, were polytetrafluoroethylene (PTFE, Teflon^®^, Wilmington, DE, USA) with density $\rho =2.2\;g/cm^{3}$, polyoxymethylene C (POM C, Delrin^®^) with density $\rho =1.41\;g/cm^{3}$, and polymethyl methacrylate (PMMA, Acrylic) with density $\rho =1.19\;g/cm^{3}$. These plastic materials have a declared density measured according to ISO 1183 and were selected to mimic the attenuation range of compact bone, spongious bone, and soft tissue, respectively. The pellets were placed side by side in a 120-degree of phase. More specifically, the low and medium-density cylinders were always in direct contact with the prosthesis. This is to simulate the worst condition where soft/fibrous tissue and trabecular bone directly interface with the metal and are therefore strongly affected by the artifacts. The highest density cylinder was placed the furthest away to simulate cortical bone. The triplets were mounted on a LEGO adjustable system (see Figure 1) with three degrees of freedom: horizontal and vertical translations in the frontal plane and rotation around the cylindrical pellet axis.

This system allows the positioning of the plastic cylinders in direct contact with the metal part of the orthopedic device. Each prosthesis was surrounded by different triplets to evaluate the artifact impact in different periprosthetic regions. To reach even small regions, like the neck part of the hip, a miniaturized triplet of pellets was assembled (the diameter of each cylinder was 5 mm). Specifically, each prosthesis was equipped with these cylindrical pellet sets (see Figure 2):-Triplets of 10-mm were placed (i) in contact with the prosthetic stem at mid length in lateral and medial position (b1, b3) and in distal position (b2); (ii) in contact with the acetabular cup in medial and lateral position (b4, b5); and (iii) in the distal region, far from metal parts (br).-Miniaturized 5-mm triplets were placed in contact with the prosthetic neck in lateral and medial positions (s1, s2).-Two HA calibrated densitometric pellets (Skyscan, Aartselaar, Belgium) were positioned (i) in contact with acetabular cup (calibrated concentrations of hydroxyapatite (HA) of 0.25 g/cm^3^); and (ii) in contact with stem in the greater trochanter region (calibrated concentrations of hydroxyapatite (HA) of 0.75 g/cm^3^).

In the following paragraphs, the term “configuration” identifies one of the different simulated scenarios with only one prosthesis (unilateral, always at the right side) or two prostheses (bilateral). We investigated four unilateral and three bilateral configurations, for a total of ten measured prostheses. Among the unilateral configurations, one was obtained by applying a cement mantle (Palacos^®^ bone cement, Heraeus Medical, Hanau, Germany) of about 3 mm to the prosthesis. In fact, cemented prostheses are frequently implanted in patients older than 80 [25]. An additional configuration is a reference, where triplets were positioned similarly, but without prostheses included. Table 1 shows the prosthetic materials for each configuration.

For bilateral configurations, for example CO+SS, the specification CO(+SS) means that artifact measurements were evaluated on CO prosthesis at the right side, while (CO+)SS means that artifact measurements were evaluated on the SS prosthesis at the left side.

### 2.3. Image Acquisition and Reconstruction

The phantom was scanned using fast-kV switching between 80 kVp and 140 kVp, 375 mA tube current, 0.7 s rotation time, 0.98435 pitch, 40 mm beam collimation, and body filter. The estimated volume computed tomography dose index (CTDIvol) provided by the DECT system for these scan parameters was 15.02 mGy. The parameters for the image reconstruction were 0.625 mm slice thickness, “Detail” kernel and adaptive statistical iterative reconstruction algorithm (ASiR) fixed at 50%. “Detail” is normally used to evaluate bony structures because it optimizes the contrast between high- and low-density materials. Virtual monochromatic imaging (VMI) was obtained at 90, 110, and 130 keV, with and without the application of the metal artifact reduction spectral (MARS) algorithm.

### 2.4. Quantitative Analysis of Metal Artifacts

To quantify the severity degree of metal artifacts and the efficacy of VMI in combination or not with MARS on metal artifact reduction, different quantities were measured in regions of interest (ROIs) placed in the pellets: CT number, noise, and SNR. For each configuration, a standardized template mask was created to obtain repeatable measurements. For each scan, the coronal slice passing by the centre of the hip prosthesis head was chosen because it could potentially be affected by a large amount of metal artifacts. The pellet ROI diameters chosen were 80% of the effective diameter. Since each scanned volume had a different display field of view (DFOV), pellet ROIs had variable diameters, which were adapted using the following equation:(1)$$\mathrm{ROI}\;\mathrm{diameter}=\frac{80\%\;\mathrm{ROI}\;\mathrm{diameter}}{\mathrm{pixel}\;\mathrm{width}}$$
where the pixel width was obtained by the DICOM file.

CT numbers were calculated by measuring the mean value of pixel intensities within the ROIs. Noise values were calculated by measuring the standard deviation of pixel intensities within the ROIs, and SNRs were obtained by dividing the CT number by the noise value of the same ROI (Appendix A). The values obtained are the means of three ROI relocations done by the reader to increase the intra-reader repeatability. From the reference configuration without prostheses, each VMI reconstruction i-th at 90, 110, 130 keV (without MARS) and 90+M, 110+M, 130+M (with MARS) provided the reference values for CT number, noise, and SNR of the pellet j-th:$$(\mathrm{CT}\;\mathrm{number}\;\mathrm{of}\;\mathrm{reference}\;\mathrm{scan}{)}_{i,j}=(N_{CT}{)}_{i,j}$$
$${\left(\mathrm{noise}\right)}_{i,j}={\left(\mathrm{standard}\;\mathrm{deviation}\;\mathrm{of}\;\mathrm{reference}\;\mathrm{scan}\right)}_{i,j}={\left(\sigma \right)}_{i,j}$$
(2)$$(\mathrm{SNR}{)}_{i,j}\;=\frac{{\left(N_{CT}\right)}_{i,j}}{{\left(\sigma \right)}_{i,j}}$$

The artifact level that affected the pellets was classified as severe or mild based on the comparison of pellet $CT_{i,j}$ numbers with the reference ${\left(N_{CT}\right)}_{i,j}$. Assuming that T is a pre-defined threshold, the criteria applied for artifact levels classification are shown in Table 2 (Appendix A).

Mild artifacts were separated in the two categories mild up / mild low, because their trends were equivalent but with opposite signs compared to the reference value, and if considered together, they could cancel each other out. For severe artifacts, just one category was chosen, considering that they only distribute below the reference value. The threshold was chosen as $T=T_{1}=850\;HU$ for low and medium-density pellets and $T=T_{2}=425\;HU\;$ for high-density pellets. The different threshold values were explained by the different pellet distances from the prosthesis since the noise is higher at the prosthesis interface and gradually decreases as we move away from the implant.

### 2.5. Statistical Analysis

CT numbers from pellets were studied to verify the dependence on the reconstruction energies and the application of MARS using a custom script (Appendix A) in MATLAB^®^ (version 2020b). To determine the effect of changing the reconstruction energies, either with or without MARS, the non-parametric two-sided Kruskal-Wallis test was used, while to determine the effect of MARS, the non-parametric two-sided Wilcoxon signed-rank test was used. In both cases, a *p*-value < 0.05 indicated the statistical significance between groups.

## 3. Results

### 3.1. Qualitative Analysis

From visual comparison, unilateral TI resulted the prosthetic configuration less affected by the presence of metal artifacts. Figure 3a shows the TI reconstruction at 130 keV for the chosen coronal plane. Hypo and hyperdense streak artifacts mostly originated from the largest sections of metallic material (proximal part of the stem, acetabular cup), probably due to the overlapping of the acetabular, neck, and stem component. The SS prosthesis was the most affected unilateral configuration, where the whole area around the stem was heavily affected by hypodense artifacts (low and medium density cylinders in stem and neck region were not visible), while hypo and hyperdense artifacts from the acetabular component were accentuated (Figure 4a). Bilateral configurations demonstrated the increase in severe artifacts due to the combined effect of the contralateral prosthesis. Figure 5 shows the same TI right prosthesis as Figure 3, but in this case coupled in the TI+SS configuration with a left SS prosthesis. For both prostheses, the area around the acetabular level presented a significant increase in artifact severity and quantity.

The qualitative impact of MARS application can be seen in Figure 3b for TI and in Figure 4b for SS. Streak artifacts disappeared on both unilateral configurations, and grey levels were more homogeneous all around the metal devices, but new MARS-related artifacts were generated. In particular, the reconstruction of the two prosthetic profiles presented a stepped appearance (Figure 3b and Figure 4b), with a significant reduction in the apparent sizes. Some halos of grey levels were still present around the periprosthetic area of the stems (Figure 3b and Figure 4b), as can be also verified in the triplets and in the PMMA cylinder with calibrated HA.

In general, triplets from b1 to b5 with pellets of 10 mm in diameter were visible and measurable, while the smaller s1–s2 triplets frequently disappeared under the artifacts. For this reason, s1–s2 triplets were excluded from the quantitative analysis.

### 3.2. Quantitative Analysis

The first analysis was on the dependence of the CT numbers with reconstruction energies and MARS application (Appendix A). Figure 6 shows the mean CT numbers from low, medium, and high-density pellets, from the b1 to b5 triplets, for all the prosthetic configurations. Low and medium-density pellets, being closer to the metal surface than the high-density pellets, were more affected by artifacts. That produced different mean values for the b1 to b5 triplets, with larger standard deviations in comparison to high-density pellets. In particular, the b1 and b3 triplets were the most affected.

The reconstruction energies did not significantly impact CT numbers in b1 to b5 triplets (Kruskal-Wallis test), either with or without MARS, with the only exception of high-density pellets with MARS, (*p* = 0.002). It should be mentioned that the large standard deviations in low-density and medium-density pellets reasonably influenced this result. On the other hand, the effect of MARS on CT numbers was statistically significant (Wilcoxon test) for low-density (*p* = 0.0004), medium-density (*p* = 0.0003), and high-density pellets (*p* = 0.007). For comparison, mean CT numbers from the reference b1 to b5 triplets obtained without prostheses did not show a significant dependence on the reconstruction energies or the application of MARS.

Starting from the same CT reference numbers, it is possible to analyse the effect of reconstruction energies and MARS on the quality and quantity of artifacts provided by the artifact level classification (severe, mild up, and mild low) (Appendix A). Figure 7, Figure 8 and Figure 9 show the mean CT number, noise, and SNR for the low, medium, and high-density pellets, respectively, for all the prosthetic configurations. Within each figure, measurements on pellets were grouped depending on the artifact level affecting the single pellet.

We found positions strongly affected by the presence of metal artifacts, such as those related to the low-density b3 pellets in CO+TI and CO+SS bilateral configurations. In some of these cases, the mean CT number measured within the ROI was −1024 ± 0 (HU), meaning that all the pixels had the lowest value on the Hounsfield scale. In these cases, the SNR cannot be calculated, and the related data were not considered in the analysis.

For the three pellet densities, the application of MARS produced CT numbers much closer to the reference values, as shown in Figure 7, Figure 8 and Figure 9. For all the densities, MARS made severe artifacts disappear, with the only exception being the 110+M reconstruction for the high-density pellets (Figure 9). For low-density, and in a lesser way for medium-density pellets, MARS also guaranteed a reduction in noise, and therefore increased values for SNR (Figure 7 and Figure 8).

Figure 10 and Figure 11 show a more specific analysis regarding the effects of VMI and MARS application for the different unilateral and bilateral prosthetic configurations. Data were reported in terms of counts of pellets affected by the different types of artifacts (severe, mild, total), as the sum of all three densities (Appendix A). Unilateral configurations presented severe artifacts (red bar) only in CO cem. and SS, both configurations at VMI 90 keV. It should be noted that only the TI configuration showed an increase in total artifact count with the introduction of MARS. For the remaining unilateral configurations, MARS made severe artifacts disappear and reduced total artifacts. The lower total artifact count in CO cem. vs. CO must be considered in light of the presence of the cement layer, which required the placement of the triplets about 3 mm farther away from the metal surface.

Bilateral configurations always presented severe artifacts on both sides, except for the TI prosthesis in TI+SS and CO+TI. In both bilateral configurations, using MARS on TI did not increase artifacts as it did in the unilateral TI configuration. For the SS and CO prostheses in bilateral configurations, MARS made severe artifacts almost completely disappear, while the effective reduction in the total number of artifacts had to be verified among the three VMI energies, and it was not always guaranteed.

By searching for the optimal combination of VMI energies and the application of MARS, it is possible to show, as in Figure 10 and Figure 11, that the conditions that generated the lowest number of total artifacts for all the prostheses in the different configurations, as summarized in Table 3. If two conditions had the same total number of artifacts, the condition presenting severe artifacts was excluded.

## 4. Discussion

In this study, a customizable phantom was defined for the in-vitro analysis of metal artifacts generated by different hip prostheses in CT scan. To simulate the in-vivo situation as much as possible, the impact of artifacts was measured on different densities which simulate compact bone (high density), spongious bone (medium density), and soft tissues (low density). A fine positioning system was implemented to accommodate all the different prosthetic models and obtain artifact measurements in direct contact with metal. These solutions provided an improvement in comparison with the previous experimental setup [7]. A method for the objective classification of the severity degree in metal artifacts was defined. Following this classification, we explored the potential of virtual monoenergetic reconstructions obtained on a dual-energy CT system. The combined effect of the software for metal artifact reduction was also studied. The analysis was applied on four unilateral and three bilateral configurations, for a total of ten measured prostheses.

As expected, unilateral configurations always presented less severe artifacts than the bilateral ones. Positions b2 (distal apex of the stem) and b5 (proximal cup) were mostly free from severe streak artifacts for all the prosthetic configurations, due to the triplets’ position with respect to the X-ray beam. The hypodense artifacts made it so the low-density pellets for the b1 (lateral stem), b3 (medial stem), and b4 (distal cup) positions were not always detectable. In the data collection from all the prostheses, the applications of different VMI energies did not significantly change the dispersed distribution of CT numbers in b1 to b5 positions. On the contrary, MARS made CT numbers much more uniform among the positions and closer to the expected reference values. In a similar way, noise and SNR did not get real advantages from the application of VMI, while MARS produced better noise and SNR values for the three densities.

The Wellenberg et al. study [7] reports that, for measurements affected by artifacts, as the monochromatic reconstruction energy rises, there is an increase in CT numbers, a noise reduction, and an increase in SNR. This is opposite to the observed energy dependency for CT numbers of bone [26] and soft tissues [27], and could be related to the presence of artifacts. In our data, it was not possible to observe these trends in a significant way for any of the three densities, for the reference, as like as for the affected measurements. This could be due to the narrow energy range in which the measurements were made in this study (90–130 keV) as Wellenberg et al. used a wider energy range (40–200 keV). Moreover, the experimental setup applied in this study generated levels of artifacts that probably overwhelmed any energy dependency of CT numbers in the affected regions. At the same time, 90–130 keV is a typical range to optimize bone contrast and artifacts in orthopedic imaging. These results on CT numbers, noise, and SNR also anticipated some measurements of metal artifact quality and quantity.

All the prostheses from unilateral and bilateral configurations presented reductions in the total number of artifacts with the application of MARS, with the only exception being the unilateral TI. This is in agreement with the literature [19], where the application of metal artifact reduction algorithm on titanium prostheses shows the generation of additional algorithm-related artifacts. The appearance of these MARS-related artifacts in particular affects the study of a thin layer of periprosthetic tissue, or whenever it is necessary to verify in fine detail, the shape, and the mechanical integrity of the prosthesis. An interesting approach to overcome this problem is from Kim et al. [16] for knee arthroplasty, where they applied virtual monochromatic reconstruction with and without MARS on the same case, to clearly identify and reset these new artifacts and improve image quality.

In our data, VMI at 110 keV always minimized the artifacts on TI prostheses for the investigated unilateral and bilateral configurations. For non-TI prostheses in all configurations, virtual monochromatic imaging alone or in combination with MARS was not determined to be a significant factor for artifact reduction. Once again, this result could be influenced by the limited range of the investigated energies (90–130 keV).

Although the effectiveness of metal artifact reduction algorithms has been established in previous studies [10,19], there are not many results on the combined application with monoenergetic reconstructions on phantoms containing hip prostheses. In particular, phantom studies have been carried out on CoCr hip prostheses [20], titanium and cobalt hip prostheses [7], and different materials with simple geometries [14,28,29]. Additionally, studies on patients have been conducted to evaluate the combined effect of monochromatic imaging and metal artifact reduction algorithms. These studies were focused on patients with different but not fully specified metallic implants [21], and on patients with CoCrMo alloy total hip prostheses [30].

In this study, an overall assessment was carried out to understand whether a standardized protocol should be used for all prosthetic materials and which configuration is less affected by artifacts among those examined. Regarding the first aim, this study suggests that different protocols should be applied depending on the prosthetic material, as recommended by the Wellenberg et al. study [15]. For titanium prostheses, the best reconstruction is obtained with VMI at 110 keV and no MARS application. Instead, in all the other prosthetic configurations, reconstructions with the MARS algorithm should be preferred, which are approximately equivalent to all energies.

In general, image quality improves with virtual monochromatic imaging at higher energies, resulting in a better representation of surrounding structures. Therefore, in principle, monochromatic energies higher than our maximum reconstruction energy value should help in reducing artifacts induced by metallic implants. However, higher energies lead to a decrease in tissue contrast. In fact, by increasing the energy, the contribution due to the photoelectric effect decreases and the contrast given by the Compton scattering increases, so the primary contrast due to the photoelectric effect disappears [31]. Actually, there are CT systems with monoenergetic reconstructions up to 200 keV, but it is reported that monoenergetic imaging in the range 140–200 keV did not show a remarkable difference in metal artifact reduction or in measured HU values [8].

In our study, it was confirmed by quantitative analysis that titanium induces fewer artifacts for unilateral and bilateral configurations. This conclusion is in agreement with the work of Wellenberg et al. [7,15] and with other studies [6,11,17]. Cemented CoCrMo stem also induced fewer artifacts, but this is probably due to the fact that pellets were located at a greater distance from the prosthesis due to the cement layer.

Of course, one limitation of this study is the applied in-vitro study on phantoms. Although it offered a replicable and customizable setup, in-vivo studies should also be performed to guarantee clinical applicability. Another limitation is the MARS-related artifacts, which should be analyzed in more detail to verify the cost-benefit ratio in using metal artifact reduction software and to identify possible countermeasures. Again, the combined analysis of in-vitro and in-vivo methods could provide a fruitful approach for MARS-related artifacts.

Aa a final remark on future developments, they must consider the application of in silico methods to simulate the artifacts for CT imaging in metal-tissue interface. In silico methods are a robust and validated method, for example, in evaluating the impact in radiotherapy treatment plans in the presence of an implanted hip prosthesis [32]. In the biomechanical field, in silico methods have also been successfully applied in evaluating contact pressure of bearing materials in hip implants [33]. However, in the specific area of metal artifacts due to hip implants in CT imaging, as far as we know from the literature, significant in silico studies are still missing. These in silico studies should be implemented for faster and less expensive results.

## 5. Conclusions

Metal artifacts affect the CT scans of patients with hip prostheses, where metal type and implant shape significantly impact the quality of periprosthetic tissue imaging. Dual-energy CT systems offer virtual monoenergetic reconstructions, with potential applications in reducing metal artifacts. The optimal choice in terms of reconstruction energy, eventually coupled with metal artifact reduction software, depends on the specific metallic implant. We developed an experimental phantom to quantify artifact severity induced by hip prostheses on different tissues (compact and spongious bone, and soft issue) by applying monoenergetic reconstructions combined with metal artifact reduction software or without it. The phantom was tested on a GE Revolution^TM^ Discovery^TM^ CT DECT system, by measuring four unilateral and three bilateral hip prosthesis configurations. The application of metal artifact reduction software on virtual monochromatic images generally improved the image quality, except for in unilateral titanium prostheses, where the application of a metal artifact reduction algorithm is not recommended. For unilateral and bilateral titanium prostheses, the reconstruction at 110 keV always minimized the artifacts. On the contrary, for cobalt and steel alloy prostheses, virtual monochromatic reconstructions alone did not show clear advantages for artifact reduction. Although affected by the limitation of an in vitro study, this study confirms that the optimal choice in terms of energy for virtual monochromatic image reconstruction, eventually coupled with metal artifact reduction software, depends on metal type and arthroplasty configuration (unilateral or bilateral). Future research will be on in-vivo imaging to evaluate the applicability of these results in a real clinical environment. Moreover, the available tools for reducing metal artifacts must be applied with care, searching for the possible countermeasures to reduce the generation of the related new artifacts. In silico methods are now offering a valuable opportunity and will be considered to proceed on this path.

## Figures and Tables

**Figure 1 diagnostics-13-00050-f001:**
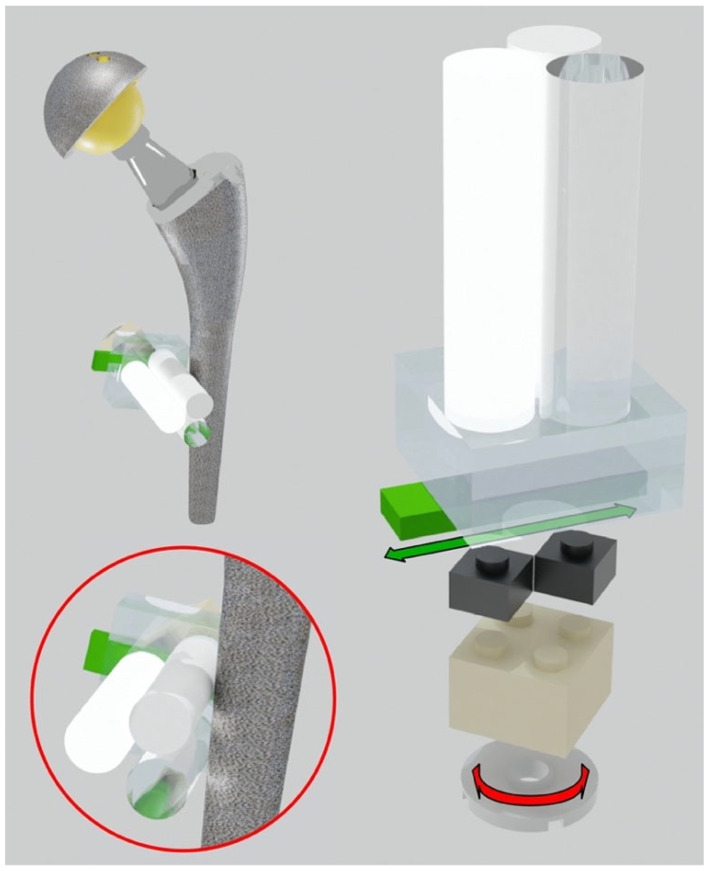
Adjustable system to put the triplets in direct contact with the hip prosthesis. Translational and rotational degrees of freedom of the triplets.

**Figure 2 diagnostics-13-00050-f002:**
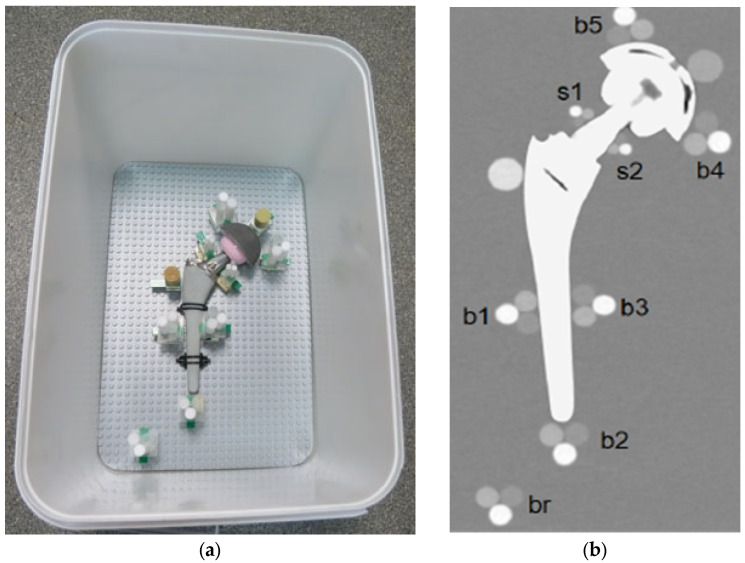
(**a**) Pellets positions around the prosthesis, TI configuration; (**b**) Coronal reconstruction.

**Figure 3 diagnostics-13-00050-f003:**
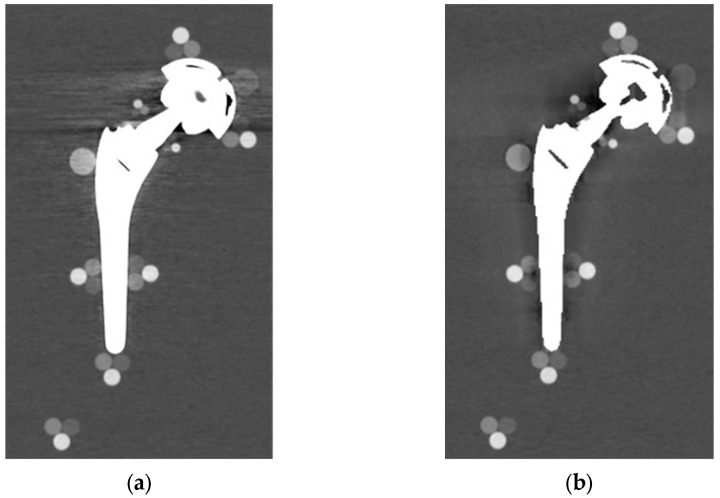
TI unilateral configuration, coronal reconstruction (bone CT visualization window WL:300, WW:1500 (**a**) virtual monochromatic imaging (VMI) reconstructed at 130 keV; and (**b**) virtual monochromatic reconstruction imaging (VMI) reconstructed at 130 keV + metal artifact reduction spectral (MARS) algorithm.

**Figure 4 diagnostics-13-00050-f004:**
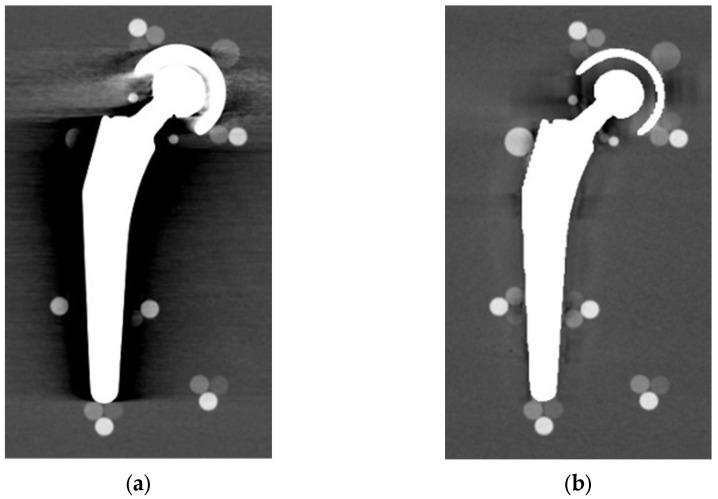
SS unilateral configuration, coronal reconstruction (bone CT visualization window WL:300, WW:1500 (**a**) VMI reconstructed at 90 keV; and (**b**) VMI reconstructed at 90 keV + MARS.

**Figure 5 diagnostics-13-00050-f005:**
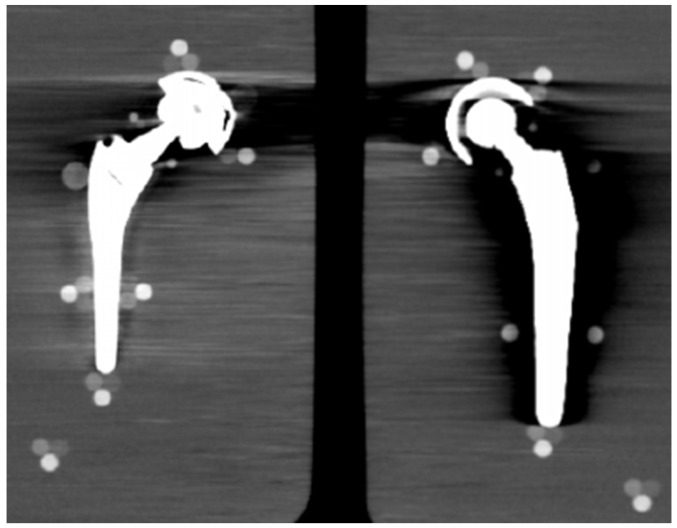
TI+SS bilateral configuration (bone CT visualization window WL:300, WW:1500). TI right prosthesis coupled with a left SS prosthesis. VMI reconstructed at 130 keV.

**Figure 6 diagnostics-13-00050-f006:**
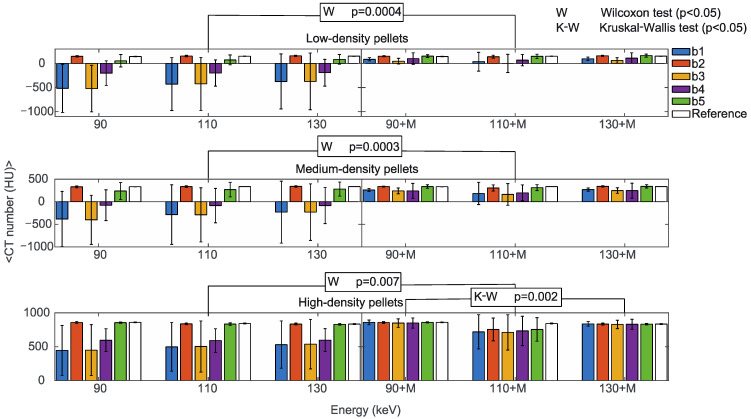
Mean CT numbers of VMI without (90, 110, 130) and with the MARS algorithm (90+M, 110+M, 130+M) for all the prosthetic configurations. Values are reported for pellets with low (upper), medium (central), and high (lower) density in positions b1, b2, b3, b4, and b5 indicated in blue, orange, yellow, purple, and green, respectively. The reference values are indicated in white. The statistically significant values of the Wilcoxon (W *p* < 0.05, to determine the effect of the MARS application) and Kruskal-Wallis (K-W *p* < 0.05, to determine the effect of changing the reconstruction energies, either with or without MARS) tests are also reported.

**Figure 7 diagnostics-13-00050-f007:**
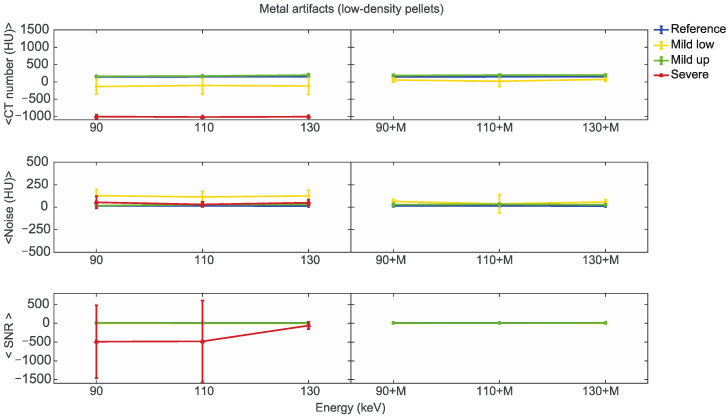
Mean CT number, noise, and SNR for low-density pellets of classified artifacts as a function of the reconstructed images, i.e., VMI without (90, 110, 130) and with the MARS algorithm (90+M, 110+M, and 130+M). The values for mild low, mild up, and severe artifact are reported with yellow, green, and red lines, respectively. The reference values are indicated with a blue line.

**Figure 8 diagnostics-13-00050-f008:**
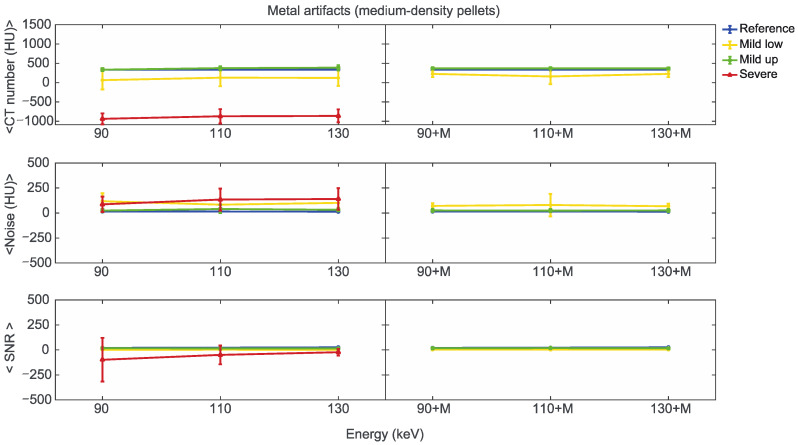
Mean CT number, noise, and SNR for medium-density pellets of classified artifacts as a function of the reconstructed images, i.e., VMI without (90, 110, 130) and with the MARS algorithm (90+M, 110+M, and 130+M). The values for mild low, mild up, and severe artifact are reported with yellow, green, and red lines, respectively. The reference values are indicated with a blue line.

**Figure 9 diagnostics-13-00050-f009:**
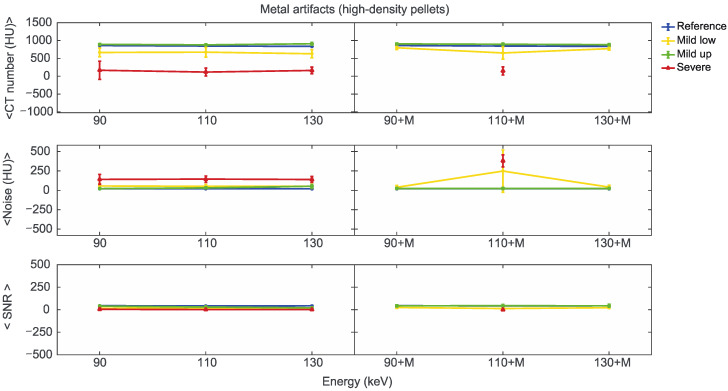
Mean CT number, noise, and SNR for high-density pellets of classified artifacts as a function of the reconstructed images, i.e., VMI without (90, 110, and 130) and with the MARS algorithm (90+M, 110+M, and 130+M). The values for mild low, mild up, and severe artifact are reported with yellow, green, and red lines, respectively. The reference values are indicated with a blue line.

**Figure 10 diagnostics-13-00050-f010:**
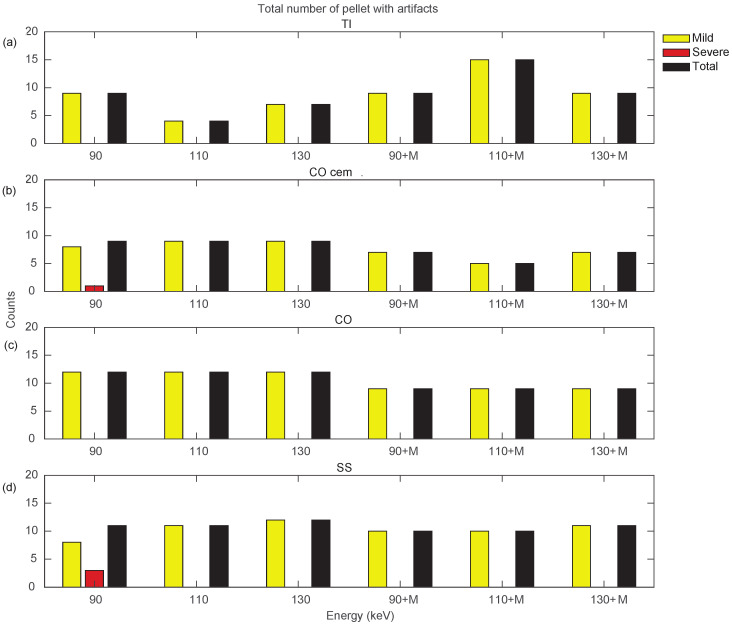
Counts for mild, severe, and total (sum of mild and severe) artifacts in prostheses from unilateral configurations, varying the reconstruction energies (VMI reconstructed without MARS, indicated as 90, 110, and 130, and VMI reconstructed with MARS, indicated as 90+M, 110+M, and 130+M). The counts of mild, severe, and total artifacts are indicated with yellow, red, and black bars, respectively. Results from TI, CO cem., CO, and SS configuration are displayed in panels (**a**–**d**), respectively.

**Figure 11 diagnostics-13-00050-f011:**
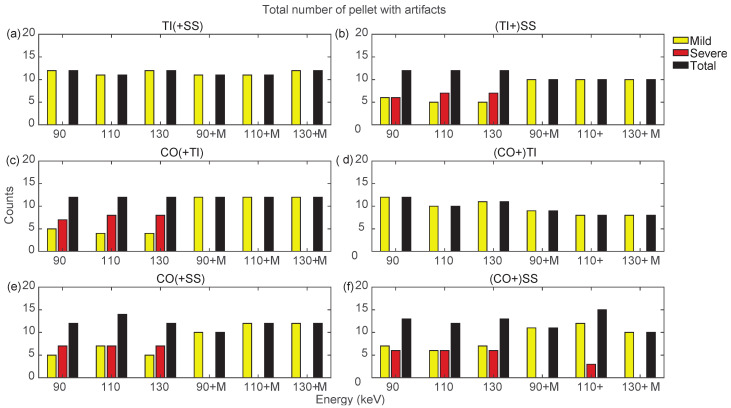
Counts for mild, severe, and total (sum of mild and severe) artifacts in prostheses from bilateral configurations, varying the reconstruction energies (VMI reconstructed without MARS, indicated as 90, 110, and 130, and VMI reconstructed with MARS, indicated as 90+M, 110+M, and 130+M). The counts of mild, severe, and total artifacts are indicated with yellow, red, and black bars, respectively. Results from TI(+SS) and (TI+)SS, CO(+TI), (CO+)TI, CO(+SS), and (CO+)SS configurations are displayed in panels (**a**–**f**), respectively.

**Table 1 diagnostics-13-00050-t001:** Prosthetic materials for unilateral and bilateral configurations. The configuration name is determined by the stem’s material.

Configuration	Right Side	Left Side
Reference	NO PROSTHESIS	NO PROSTHESIS
TI	Stem: Ti6Al4V Cup: Ti6Al4V Insert: zirconia toughened alumina Head: zirconia toughened alumina	NO PROSTHESIS
CO cem.	Stem: CoCrMo cemented Cup: Ti6Al4V Insert: UHWMPE Head: CoCrMo	NO PROSTHESIS
CO	Stem: CoCrMo Cup: stainless steel Insert: UHWMPE Head: CoCrMo	NO PROSTHESIS
SS	Stem: stainless steel Cup: stainless steel Insert: UHWMPE Head: stainless steel	NO PROSTHESIS
TI+SS	Stem: Ti6Al4V Cup: Ti6Al4V Insert: zirconia toughened alumina Head: zirconia toughened alumina	Stem: stainless steel Cup: stainless steel Insert: UHWMPE Head: stainless steel
CO+SS	Stem: CoCrMo Cup: stainless steel Insert: UHWMPE Head: CoCrMo	Stem: stainless steel Cup: stainless steel Insert: UHWMPE Head: stainless steel
CO+TI	Stem: CoCrMo Cup: stainless steel Insert: UHWMPE Head: CoCrMo	Stem: Ti6Al4V Cup: Ti6Al4V Insert: zirconia toughened alumina Head: zirconia toughened alumina

ISO references for prosthetic materials: Ti6Al4V/ISO 5832-3; CoCrMo/ISO 5832-4; Ceramic (zirconia toughened alumina)/ISO 6474-2; stainless steel/ISO 5832-9; UHWMPE (Ultra-high-molecular-weight polyethylene)/ISO 5834-2.

**Table 2 diagnostics-13-00050-t002:** Criteria for pellets classification in relation to the level of artifact.

${\mathrm{CT}}_{i,j}$ Number from Pellet (HU)	Artifact Level on Pellet
(*N_CT_* *−* *σ*)_i,j_ < *CT*_i,j_ < (*N_CT_* + *σ*)_i,j_	Not affected
(*N_CT_* + *σ*)_i,j_ < *CT*_i,j_ < (*N_CT_* + *σ*)_i,j_ + *T*	Mild up
(*N_CT_* − *σ*)_i,j_ − *T* < *CT*_i,j_ < (*N_CT_* − *σ*)_i,j_	Mild low
*CT*_i,j_ < (*N_CT_* − *σ*)_i,j_ − *T*	Severe

**Table 3 diagnostics-13-00050-t003:** Suggested combination of VMI and MARS to minimize metal artifacts for implant-specific CT imaging. The check mark ✓ indicates the conditions that generated the lowest number of total artifatcs.

	Virtual Monochromatic Imaging (VMI)			Virtual Monochromatic Imaging + Metal Artifact Reduction Spectral Algorithm (VMI + MARS)		
Prosthesis	90 keV	110 keV	130 keV	90 keV + MARS	110 keV + MARS	130 keV + MARS
TI		✓				
CO cem.					✓	
CO				✓	✓	✓
SS				✓	✓	
TI(+SS)		✓		✓	✓	
(TI+)SS				✓	✓	✓
(CO+)TI					✓	✓
CO(+TI)				✓	✓	✓
CO(+SS)				✓		
(CO+)SS						✓

## Data Availability

The data presented in this study are available in Appendix A.

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
