# Peer review of "Dual-Energy Computed Tomography Applications to Reduce Metal Artifacts in Hip Prostheses: A Phantom Study"

_diagnostics, 2022, doi:10.3390/diagnostics13010050_

Round 1

Reviewer 1 Report

The authors present a study about metal artifact assessment in different configurations of prostheses. A reproducible in-vitro setup is used with a DECT setup. The effect of MA on different tissues (bone, spongious bone and soft tissue) is quantified by the mean difference of HUs in homogeneous ROIs. Virtual monoenergetic images in combination with MAR algorithms are tested for their performance on accurate HU quantification. Suggestions for the most beneficial reconstruction pipelines in several configurations of prostheses are given in the discussion and conclusion section.

Abstract: Please include the main results in the abstract. Define severity level (HU deviation from reference).

2.1.: Line 119 "to overstress the artifact evaluation" please rephrase.

2.2.: Why are prostheses not investigated in bilateral combinations TI+TI, SS+SS and CO+CO?

2.3.: Please indicate the dose level of CT acquisitions.

2.4.: Line 216 "standard deviation from reference" is misleading. -> "standard deviation of reference scan"

Line 239: Are the thresholds given in HU?

3.2.: Why do different energy levels not lead to changing HU values in reference triplets? What about the energy dependency of HU values?

Figure 5 and further: Please increase font size.

Line 319: Is the CT number a mean value over all prosthetic configurations? (also in line 329)

Discussion: Line 428: Why should the CT number increase with increasing energy levels? See the energy dependency of soft tissue (https://physics.nist.gov/PhysRefData/XrayMassCoef/ComTab/tissue.html) and bone (https://physics.nist.gov/PhysRefData/XrayMassCoef/ComTab/bone.html).

Reviewer 2 Report

The abstract should be broadened to give additional quantitative results.

1.      Please add the abstract's "take-home" message, the current form was insufficient.

2.      Sort the keywords according to alphabetical order.

3.      The Reviewer do not see the novel in the present article. My examination revealed that several similar previous publications appear to appropriately address the issues you have brought up in the current submission. Please emphasize it more advance in the introduction section if there are any more truly something really new.

4.      Previous study related needs to explain in the introduction section consisting of their work, their novelty, and their limitations to show the research gaps that intend to be filled in the present study.

5.      Why present study only performs in vitro? Not in silico and in vivo? Solid explanation is needed.

6.      Since the present study performs metallic implant investigation in vitro, potential study in silico perspective needs to be explained that offer several advantages such as lower cost and faster results. It is a vital topic that authors must provide in the introduction and/or discussion section. Additionally, the MDPI's suggested reverence should be taken to substantiate this explanation as follows: Jamari, J.; Ammarullah, M. I.; Santoso, G.; Sugiharto, S.; Supriyono, T.; van der Heide, E. In Silico Contact Pressure of Metal-on-Metal Total Hip Implant with Different Materials Subjected to Gait Loading. Metals (Basel). 2022, 12, 1241. https://doi.org/10.3390/met12081241

7.      Rather than relying just on the predominate text as it already exists, the authors could incorporate more illustrations as figures in the materials and methods section that illustrate the workflow of the current study.

8.      Additional information about tools used, such as the maker, country, and specification, should be included.

9.      The inaccuracy and tolerance of the experimental equipment used in this inquiry are critical details that must be included in the article.

10.   Findings must be compared to similar past research.

11.   Overall, discussion in the present article is extremely poor. The Authors must extend their discussion and make a comprehensive explanation. Just not simply mention the results with brief explanation.

12.   Extend the limitations of present study in line 481-483. It is should be more.

13.   The conclusion section needs to explain further research.

14.   The authors should give additional references from the five-years back. MDPI reference is strongly recommended.

15.   The authors occasionally created paragraphs in the entire document that were just one or two phrases long, which made the explanation difficult to understand. To make their explanation into a longer, more thorough paragraph, the authors should expand it. It is advised to use at least three sentences in a paragraph, with one serving as the primary sentence and the others as supporting phrases.

16.   The authors were encouraged to proofread their work due to grammatical problems and linguistic style.

17.   After peer review, it is encouraged that a graphical abstract be included in the submission.

Round 2

Reviewer 2 Report

Well done revised by The Authors.